# Yes-Associated Protein Is Required for ZO-1-Mediated Tight-Junction Integrity and Cell Migration in E-Cadherin-Restored AGS Gastric Cancer Cells

**DOI:** 10.3390/biomedicines9091264

**Published:** 2021-09-18

**Authors:** Seon-Young Kim, Song-Yi Park, Hwan-Seok Jang, Yong-Doo Park, Sun-Ho Kee

**Affiliations:** 1Department of Microbiology, College of Medicine, Korea University, Seoul 02841, Korea; ksy0817@korea.ac.kr (S.-Y.K.); songyip3507@korea.ac.kr (S.-Y.P.); 2Department of Biomedical Sciences, College of Medicine, Korea University, Seoul 02841, Korea; kevin14@korea.ac.kr (H.-S.J.); ydpark67@korea.ac.kr (Y.-D.P.)

**Keywords:** E-cadherin, ZO-1, YAP, angiomotin, tight junction, cell migration

## Abstract

Yes-associated protein (YAP) regulates numerous cellular homeostasis processes and malignant transformation. We found that YAP influences ZO-1-mediated cell migration using E-cadherin-restored EC96 cells derived from gastric malignant AGS cells. Ectopic expression of E-cadherin enhanced straightforward migration of cells, in comparison to the meandering movement of parental AGS cells. In EC96 cells, YAP and ZO-1 expression increased but nuclear YAP levels and activity were reduced. Nuclear factor-κB (NF-κB) mediated the increase in ZO-1 expression, possibly stabilizing cytoplasmic YAP post-translationally. Downregulation of YAP expression using siYAP RNA or stable knock-down inhibited straightforward cell migration by fragmenting ZO-1 containing tight junctions (TJs) but not adherens junctions, implying involvement of YAP in ZO-1-mediated cell migration. The association of YAP with ZO-1 was mediated by angiomotin (AMOT) because downregulation of AMOT dissociated YAP from ZO-1 and reduced cell migration. E-cadherin restoration in malignant cancer cells induced NF-κB signaling to enhance ZO-1 expression and subsequently stabilize YAP. At high expression levels, YAP associates with ZO-1 via AMOT at TJs, influencing ZO-1-mediated cell migration and maintaining TJ integrity.

## 1. Introduction

The Hippo signaling pathway and its major effector protein, Yes-associated protein (YAP), are evolutionarily conserved regulators of cell growth, organ size, and tissue homeostasis in a variety of species from *Drosophila* to mammals [1,2]. The Hippo pathway proceeds to the kinase cascade, MST1/2 and LATS1/2, resulting in phosphorylation of YAP, which negatively regulates YAP nuclear translocation and activity including transcription of tumorigenesis-associated target genes such as connective tissue growth factor (CTGF) and cysteine-rich angiogenic protein 61 (Cyr61) [3,4]. Dysregulation of the Hippo pathway and YAP overactivation promotes cell proliferation and resistance to death and is associated with various cancers [4]. Additionally, shuttling of cytoplasmic YAP into the nucleus is critical for tumorigenesis and metastasis [5]. In cytoplasm, YAP is phosphorylated and subjected to ubiquitination and degradation [6]. YAP overexpression in the cytoplasm and nucleus has been observed in various malignant cancers [7,8,9]. In *Drosophila*, Ex interacts with Yorkie (Yki), an ortholog of YAP, promoting retention of Yki in the cytoplasm [10]. In mammalian cells, angiomotin (AMOT) induces cytoplasmic retention of YAP by direct binding [11]. Mechanisms other than the Hippo pathway also regulate YAP activity. Regulation of post-translational modifications such as ubiquitination, sumoylation, and acetylation controls the cytoplasmic level of YAP and its nuclear translocation [6]. Additionally, YAP activity can be affected by cellular contexts such as cell-junction formation and cell polarity [12].

YAP activity is affected by cell–cell junction formation, such as adherens junctions (AJs) and tight junctions (TJs) and establishment of cellular polarity at apical–basal junctions [13]. Disruption of TJs or AJs in cultured mammalian cells induces YAP nuclear localization and target gene expression [12,14]. E-cadherin, a major AJ component, controls the cell-density-dependent subcellular localization of YAP. Knockdown of β-catenin in densely cultured MCF10A cells decreases phosphorylation of the S127 residue of YAP and its nuclear accumulation [15]. β-catenin, a major effector of the Wnt signaling pathway and a component of AJs, binds with YAP directly and mediates cross-regulation between the Hippo and Wnt signaling pathways, regulating cell phenotypes including proliferation [16]. Several components of TJs regulate Hippo and YAP activities [12]. Claudin-18 (CLDN18) interacts with YAP and co-localizes at cell–cell contacts, and loss of CLDN18 suppresses the interaction of YAP with LATS1/2 in alveolar epithelial cells [17]. ZO-2 induces YAP nuclear localization [12,18], whereas ZO-1 represses the activity of TAZ, a paralog of YAP [19]. Beyond the classical barrier functions of TJ structures [20], downregulation of TJ proteins such as claudins, occludin, and ZO-1 reduces metastatic features, including cell migration [21,22,23]. Angiomotin (AMOT) family proteins interact with YAP and multiple components of TJs and AJs, such as β-catenin and ZO-1 [24,25]. The scaffolding functions of AMOT lead to sequestration of YAP to TJs, reducing nuclear YAP activity and maintaining TJ integrity and epithelial cell polarity [11,26].

E-cadherin restoration in gastric cancer cells leads to acquisition of malignant phenotypes such as enhanced cell proliferation and higher energy production with increased glucose uptake [27]. E-cadherin, a negative regulator of cell invasion and epithelial-mesenchymal transition (EMT), is important for malignant phenotypes and cell survival within metastatic sites [28,29]. Additionally, TJ structure and components play roles in cancer progression and metastasis beyond its classical structural functions in the epidermis [30]. Herein, we observed that E-cadherin restoration elevated expression of ZO-1 and YAP, which was accompanied by increased cell migration in the context of reduced YAP nuclear accumulation and activity. To identify the underlying mechanism, we investigated the effect of YAP on ZO-1-mediated TJ structures and cell migration.

## 2. Materials and Methods

### 2.1. Cell Culture and Transfection

The human gastric cancer cell line AGS was purchased from Korea Cell Line Bank (Seoul, Korea) in 2003. EC96 cells were cultured in Dulbecco’s Modified Eagle’s Medium (Serana, Pessin, Germany) containing 10% (*v*/*v*) fetal bovine serum (FBS; Biotechnics Research Inc, Lake Forest, CA, USA) and 0.5% Penicillin/streptomycin (Lonza, Walkersville, MD, USA). The establishment of EC96 was described previously [27]. All cells were incubated in an incubator with 5% CO_2_ at 37°C. For the transfection of siRNA oligonucleotide, cells which reached 60–70% confluence were transfected with siRNA using Lipofectamine-RNAiMax (Invitrogen, Carlsbad, CA, USA) according to the manufacturer’s protocol. Sequences of siRNAs are listed in Appendix A.

### 2.2. Lentiviral Infection and Generation of Stable Cell Lines

To generate YAP and ZO-1 knockdown cells (YAP KD and ZO-1 KD), shYAP and shZO-1 RNA in pLKO.1 lentiviral vectors were purchased from Sigma (St. Louis, MO, USA). For packaging lentivirus, 293T cells were co-transfected with pLKO.1 constructs and packaging plasmids (psPAX2, pMD2.G, and VSV-G). The media containing virus released from 293T cells were collected, filtered, and used to infect AGS and EC96 cells. Infected cells were maintained in culture medium containing 1 μg/mL puromycin (InvivoGen, San Diego, CA, USA). Puromycin-resistant colonies were isolated. The gene silencing was confirmed by immunoblotting.

### 2.3. Cell Migration Assay Using Scratch Method

Cell migration was induced by scratching the cells at about 90% confluence with a pipette tip and replacing them with fresh 10% FBS containing medium. After scratching, cells were photographed at indicated time points using a Dino-Eye Digital Eyepiece Camera (Dino-Lite, Taiwan), and the change in scratch area was measured using Image J software (NIH, Bethesda, MD, USA).

### 2.4. Cell Migration Assay Using Cell Island Patterning

Fabrication of Polydimethylsiloxane (PDMS) stencils and micropatterning of cell islands has been described previously [31]. The patterned cell islands were incubated to grow and bright-field images for cells were acquired every 10 min for up to 9 h using a JuLI stage live cell imaging system (NanoEnTek, Seoul, Korea) with a 4× magnification objective lens (Olympus, Tokyo, Japan) housed within an incubator. The acquired bright-field cell images were numerically transformed and quantitatively analyzed using custom codes written (MathWorks Inc., Carlsbad, CA, USA) to calculate cell velocities and trajectories according to previous descriptions [31].

### 2.5. RNA Extraction and Quantitative Real-Time PCR

TRIzol reagent (Life technologies, Carlsbad, CA, USA) was used to isolate total RNA from cells. Quantitative real-time PCR was performed using SYBR Green PCR Master Mix (Applied Biosystems, Foster City, CA, USA). cDNAs were subjected to quantitative real-time PCR on the 7500 Fast Real-time PCR System. Sequences of quantitative RT-PCR analysis primers are listed in Appendix A.

### 2.6. Subcellular Fractionation

Cells were harvested by scraping from dishes. Cells were lysed with hypotonic lysis buffer (10 mM HEPES pH 7.9, 10 mM KCl, 0.1 mM EDTA, 1 mM DTT, and 1X protease inhibitor), incubated on ice for 15 min, and 25 μL 10% NP-40 was added followed by a centrifugation at 13,000 rpm for 5 min at 4 °C. The supernatant was kept as cytoplasmic extract and the pellet was resuspended hypotonic buffer. This washing step was repeated three times. Pellets were resuspended in nuclear lysis buffer (20 mM HEPES pH 7.9, 0.4 M NaCl, 1 mM EDTA, 1 mM DTT, 10% NP-40) and then vortexed vigorously for 10 min. The samples were centrifuged at 13,000 rpm at 4 °C for 10 min and the supernatant was kept as nuclear extracts.

### 2.7. Immunoblotting Immunofluorescence, and Immunoprecipitation

Immunoblotting, immunofluorescence (IF) and immunoprecipitation (IP) assays were performed according to previous description [27]. For enhanced visualization of junctional proteins in an IF assay, cells were pretreated with 0.4% Triton X-100 and 0.4% paraformaldehyde in phosphate-buffered saline prior to fixation. The following antibodies were used: E-cadherin, β-catenin, Lamin A/C (BD Biosciences, San Jose, CA, USA), ZO-1 (Invitrogen), YAP, Angiomotin (AMOT), NF-κB p65, β-actin (Santa Cruz Biotechnology), p-YAP (Ser127), and p-YAP (Ser397) (Cell Signaling Technology, Danvers, MA, USA). Verteporfin (R&D Systems, Minneapolis, MN, USA) and Bay11-7082 (InvivoGen) were dissolved in DMSO and used.

### 2.8. Statistical Analysis

All experiments were repeated 3 times in each group. Statistical analysis was performed by the Student *t* test (two-tailed). All data are presented as the mean ± standard error. Differences with *p* < 0.05 were considered statistically significant. A *p* value: * < 0.05, ** < 0.01, *** < 0.001.

## 3. Results

### 3.1. E-cadherin Expression Enhances Cell Migration and Expression of YAP and ZO-1

Previously, we established EC96 cells by re-introducing E-cadherin to AGS gastric cancer cells and observed acquisition of more malignant phenotypes by activation of the NF-κB signaling pathway [27]. The migration rate of EC96 cells was approximately 1.6-fold higher than that of AGS cells (Figure 1A). By immunoblotting, EC96 cells showed a higher expression of E-cadherin, ZO-1, and phosphorylated YAP at serine-127 and -397 compared to AGS cells (Figure 1B). qRT-PCR analysis revealed that transcripts of ZO-1, but not YAP, were increased in EC96 cells (Figure 1C), implying that the increase in YAP expression was induced by a post-translational regulatory mechanism.

In EC96 cells, increased YAP expression was accompanied by its phosphorylation at serine-127 (Figure 1B), which inhibits YAP nuclear accumulation and transcriptional activity [23]. A subcellular fractionation analysis revealed that YAP was more accumulated in cytoplasm because of reduced nuclear translocation in EC96 cells in comparison to AGS cells (Figure 1D). The transcript levels of YAP target genes, such as CTGF and Cyr61, were also reduced in EC96 cells (Figure 1E), indicating reduced YAP activity. Phosphorylated YAP at serine-127 and -397 accumulated at cell–cell junctions (Appendix A), although phosphorylation at serine-397 is linked to ubiquitination and degradation [32]. Consistent with our previous results [27], NF-κB signaling appeared to be elevated in EC96 cells as high levels of nuclear NF-κB was observed (Figure 1F). Furthermore, inhibition of NF-κB signaling using Bay11-7082 reduced cell migration as well as expressions of ZO-1 and YAP in EC96 cells (Appendix A), indicating involvement of NF-κB signaling.

### 3.2. YAP and ZO-1 Participate in Regulation of Cell Migration

Treatment with verteporfin (VP), which disrupts the YAP-TEAD interaction, reduced cell migration in EC96 cells (Figure 2A). Transfection of siYAP RNAs reduced migration of AGS and EC96 cells and decreased ZO-1 expression but did not affect that of β-catenin and E-cadherin (Figure 2B,C). Considering its involvement in cell migration [22,31], we analyzed the effect of ZO-1 on cell migration using ZO-1 knock-down cells (ZO-1 KD1 and KD2 cells); cell migration was reduced in all KD cells (Figure 2D,E), implicating YAP and ZO-1 in regulation of cell migration. Additionally, ZO-1 KD cells showed reduced YAP expression (Figure 2D). YAP and ZO-1 expression was increased in a confluence-dependent manner (Figure 2F). These results imply reciprocal regulation of ZO-1 and YAP. A qRT-PCR analysis showed that siYAP RNA transfection and ZO-1 KD did not affect the transcript levels of ZO-1 and YAP, respectively, implying reciprocal stabilizing effects. YAP KD reduced ZO-1 transcription (Appendix A), implying that long-term reduction of YAP expression reduces ZO-1 expression.

### 3.3. ZO-1 Interacts with YAP at Cell Membranes

Our results led us to speculate that YAP and ZO-1 cross-regulate, thus influencing cell migration. An IP analysis revealed that YAP or ZO-1 co-precipitated with ZO-1 or YAP, respectively (Figure 3A). β-catenin and ZO-1 were not detected in precipitates of anti-ZO-1 and anti-β-catenin antibodies, respectively (Figure 3A). IF showed that YAP co-localized with ZO-1 at cell junctions, and to a greater extent in EC96 cells (Figure 3B, arrows). β-catenin was not detected at cell junctions, where YAP was located in AGS cells (Figure 3C, arrows), but YAP, ZO-1, and β-catenin were present at cell junctions of EC96 cells (Figure 3B,C, arrows). YAP interacted with β-catenin (Figure 3A) in the cytoplasm because co-localization of YAP and β-catenin was not detected at cell junctions of AGS cells (Figure 3C). An IF analysis of EC96 ZO-1 KD cells showed that ZO-1 and YAP were decreased at cell junctions (Figure 3D), but β-catenin was present at cell junctions (Figure 3E, arrowheads), implying that the association of YAP with ZO-1 at cell junctions is independent of E-cadherin and β-catenin. Subcellular fractionation showed that nuclear YAP was slightly increased in ZO-1 KD AGS and EC96 cells (Figure 3F). Therefore, ZO-1 expression induces membrane sequestration of YAP and prevents its nuclear translocation.

### 3.4. AMOT Links YAP to ZO-1 at Tight Junctions

AMOT interacts with YAP and localizes to TJs [11], implying a role for AMOT in localization of YAP at cell membranes, especially TJs. AMOT expression was higher in EC96 cells than in AGS cells, similar to YAP and ZO-1 (Figure 4A). As expected, IP analysis using anti-ZO-1 or anti-YAP antibodies showed co-precipitation of AMOT with ZO-1 and YAP and co-localization of AMOT with ZO-1 and YAP (Appendix A). AGS and EC96 cells were transfected with siAMOT RNA and showed reduction of AMOT, YAP, and ZO-1 expression (Figure 4B); similarly, AMOT expression was decreased in ZO-1 KD and YAP KD cells (Appendix A). Additionally, migration of AGS and EC96 cells was significantly inhibited (Figure 4C), implying that AMOT is involved in the regulation of cell migration.

### 3.5. YAP Regulates ZO-1-Mediated Tight Junction Structures

To assess the effect of YAP on TJ structures, an IF analysis of siYAP-transfected cells was performed using an anti-ZO-1 antibody. Upon transfection, a continuous linear pattern of ZO-1 was transformed to a fragmented or frequently punctuated pattern at cell junctions of AGS and EC96 cells (Figure 5A). To confirm these results, YAP expression was restored by transfection of a YAP overexpression plasmid (Figure 5B). YAP restoration restored fragmented ZO-1 staining to continuous linear TJ structures in stable YAP KD cells (Figure 5C). β-catenin expression was not affected in YAP KD cells, implying that YAP regulation of ZO-1 mediated TJ structures is independent of AJ structures (Figure 5D). The failure of co-localization of ZO-1 with β-catenin supports this result (Figure 5D). These results imply that YAP membrane localization is important for maintaining the integrity of ZO-1-containing TJ structures, which might facilitate EC96 cell migration. In addition, siAMOT RNA transfection induced fragmentation of ZO-1-containing TJ structures and dissociation of YAP from ZO-1 (Figure 5E, arrows), implying an AMOT-mediated linkage between YAP and ZO-1, which is crucial for the maintenance of TJ integrity.

### 3.6. YAP Is Required for Straightforward Movement of EC96 Cells

To verify the involvement of YAP in regulation of cell migration, we used a cell island model [31] in which the number and density of clustered cells were controlled using a pattern of homogeneous size and shape. The cell island started at 1 mm in diameter and expansion by isotropic free-edge migration was monitored for 9 h. The EC96 cell island expanded more widely than did the AGS cell island (Figure 6A,B). The enhanced expansion of EC96 cells was decreased by ZO-1 and YAP knock-down (EC96 ZO-1 KD and YAP KD cells). The area expansion of EC96 appeared to increase slightly even after the data were corrected by initial cell number (Appendix A). To verify if migration of EC96 indeed elevated, cellular trajectories were analyzed. The trajectory analysis showed that AGS cells acquired straightforward movement upon E-cadherin restoration, which was mediated by ZO-1 [31]. The trajectory of individual EC96 cells was decreased by YAP knock-down, and the effect was more evident in cells at the colony periphery (Figure 6C). Additionally, trajectory analysis showed EC96 cells located at the colony marginal border moved faster in a straightforward pattern, but this was abolished by YAP knockdown (Figure 6D). These results imply that together with ZO-1, YAP is involved in regulation of cell movement.

## 4. Discussion

E-cadherin plays an important role in cell–cell adhesion and regulates cellular processes including migration [33]. Development and reduction of E-cadherin is essential to the EMT, which is associated with cell migration, invasion, and metastasis [34]. Cells that have metastasized through the EMT process settle in other organs and form new tumors, where E-cadherin is re-expressed [35,36]. Re-introduction of E-cadherin enhances cell proliferation and ATP production by activating NF-κB signaling [27]. The E-cadherin-mediated AJ structure in the cell membrane sequestrates β-catenin, a key component of Wnt signaling, inhibiting Wnt signaling [37]. In addition, β-catenin binds to YAP and regulates its activity by sequestrating YAP in the cytoplasm [16], inhibiting malignant transformation. Therefore, reduction of Wnt signaling and YAP activity in EC96 cells is unsurprising. However, proliferation [27] and migration were increased in EC96 cells compared to AGS cells (Figure 1A). These contradictory results may be explained by a compensatory increase in NF-κB signaling activity (Figure 1 and Appendix A). To identify underlying mechanisms other than NF-κB signaling, we evaluated the role of cytoplasmic YAP expression in the malignant phenotypes of EC96 cells.

EC96 cells showed increased expression of YAP with phosphorylation at Ser-127 and -397 (Figure 1B) and decreased nuclear YAP expression (Figure 1D). YAP has several phosphorylation consensus motifs of LATs, in which phosphorylation of S127 results in 14-3-3 binding and cytoplasmic retention of YAP leads to its ubiquitination and degradation [32]. In EC96 cells, phosphorylated YAP localized to the cell membrane instead of being degraded in the cytoplasm, implying retention of YAP at membranes (Figure 1 and Appendix A). The increased YAP expression at membranes implies additional roles, because the role of membrane YAP is unknown. YAP inhibition using verteporfin and siYAP RNA transfection reduced migration of AGS and EC96 cells (Figure 2A,C), implying that YAP is required for cell migration. Because continuous linear staining of ZO-1 was transformed to fragments in YAP KD EC96 cells (Figure 5), YAP might affect the integrity of TJ structures.

YAP and ZO-1 expression increases in a confluence-dependent manner [15,38], consistent with our results (Figure 2F). Downregulation of any of YAP, ZO-1, and AMOT also reduced the expression of the other two factors (Figure 2 and Figure 4), implying mutual regulation. However, the transcript levels of ZO-1 and YAP were not affected by siYAP RNA transfection and ZO-1 KD, respectively (Appendix A), suggesting of post-transcriptional regulation.

AMOT interacts with YAP [11] as well as multiple TJ components [25], and we found clustering of ZO-1, YAP, and AMOT (Figure 3, Figure 5 and Appendix A). AMOT is important for maintaining TJ integrity and epithelial cell polarity [25]. AMOT interacts with YAP, inhibiting YAP nuclear translocation [11]. Therefore, the association of AMOT with ZO-1 may recruit YAP to membranes, reducing nuclear YAP. IF showed that transfection of siAMOT RNA induced separation of YAP from ZO-1 at the membrane of AGS and EC96 cells, implying a scaffolding effect of AMOT for YAP and ZO-1 (Figure 5E). Additionally, downregulation of AMOT dissociated YAP from ZO-1 (Figure 5) and reduced expression of YAP and ZO-1 (Figure 4), implying that clustering of YAP, ZO-1, and AMOT is important for their stability and expression. This association at the membrane might enhance ZO-1 mediated TJ integrity and cell migration because downregulation of the three proteins reduced cell migration (Figure 2, Figure 4 and Figure 6).

ZO-1 is a scaffolding component in the assembly of TJs, and functions as a barrier to control the movement of electrolytes and water [20]. In addition, ZO-1 inhibits tumor metastasis and regulates cell proliferation and migration [22,39]. For example, ZO-1 RNAi-mediated knockdown largely abrogated cell movement following wounding in COS-7 cells [22]. In melanoma cells, ZO-1 exists at heterologous junctions between melanoma cells and fibroblasts, implying involvement in melanoma invasiveness [40]. Additionally, ZO-1 is implicated in directional movement of EC96 cells [31]. Similar to ZO-1 KD cells, YAP KD reduced straightforward movement of EC96 cells, especially boundary cells (Figure 6), and reduced their migration (Figure 2). Therefore, YAP may regulate ZO-1-mediated cell migration.

YAP interacts with β-catenin [24] and influences Wnt signaling activity positively or negatively [16]. ZO-1 promotes establishment of AJ structures by interacting with α-catenin and the actin cytoskeleton [41]. E-cadherin restoration increased ZO-1 expression, implying that E-cadherin acts upstream to regulate YAP and ZO-1 expression (Figure 1). However, YAP-mediated regulation of the integrity of ZO-1-containing TJ structures may be independent of E-cadherin/β-catenin, because ZO-1 failed to co-localize with β-catenin at the membrane in EC96 cells, and β-catenin expression was maintained but ZO-1 expression was significantly reduced in YAP KD EC96 cells (Figure 5D). The discrepancy between observations may result from differences in the cellular context. In addition, these cell culture-based experiments required to expand in vivo experiments for further verification.

In conclusion, E-cadherin restoration in malignant cancer cells induces NF-κB signaling to compensate for suppression of YAP and Wnt signaling, increasing ZO-1 expression and stabilizing YAP expression. YAP associates with ZO-1 via AMOT at TJs, influencing ZO-1-mediated cell migration and maintaining TJ integrity.

## Figures and Tables

**Figure 1 biomedicines-09-01264-f001:**
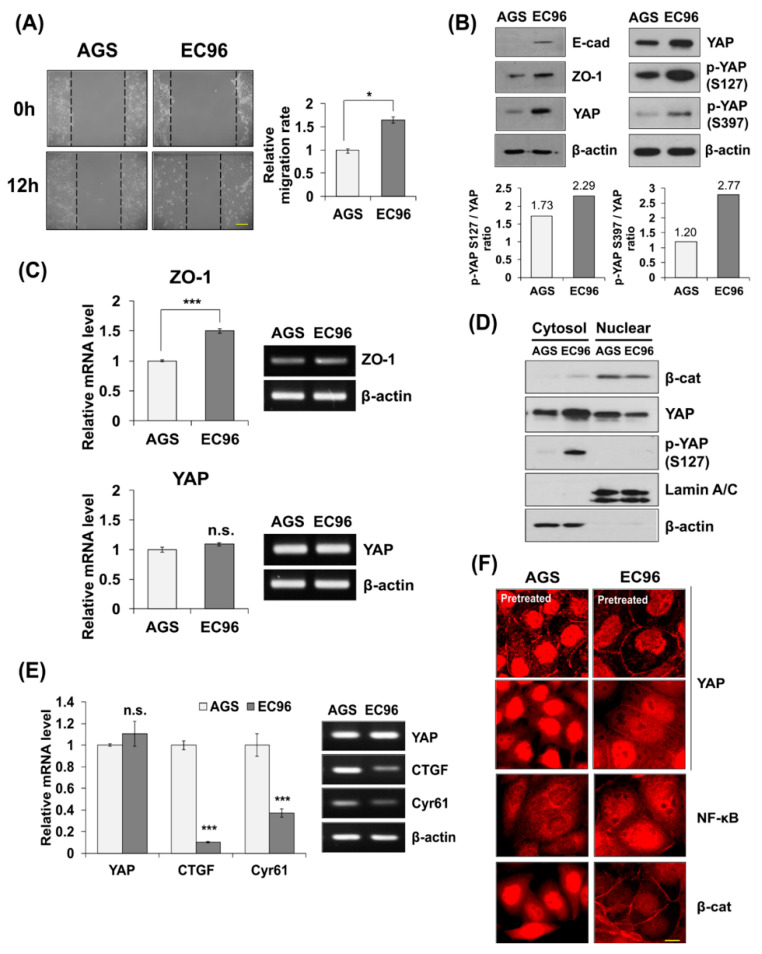
E-cadherin expression increases YAP expression and inhibits nuclear YAP translocation. (**A**) AGS and EC96 cells were subjected to cell migration assay. Cells were incubated for 12 h and wound gaps were measured. Results are means ± SD of three experiments. * *p* < 0.05. Scale bar = 100 μm. (**B**) AGS and EC96 cells were subjected to immunoblot analysis for the indicated proteins. Densitometry analysis was performed to evaluate p-YAP S127/YAP and p-YAP S397/YAP ratios. (**C**–**F**) AGS and EC96 cells were subjected to qRT-PCR analysis for ZO-1 and YAP transcripts (**C**), subcellular fractionation to analyze localization of YAP (**D**), qRT-PCR of YAP target genes, such as CTGF and Cyr61 (**E**), and IF analyses for YAP, NF-κB, and β-catenin (β-cat) (**F**), n.s. = not significant and *** *p* < 0.001 in (**C**) or (**E**). Scale bar = 10 μm in (**F**).

**Figure 2 biomedicines-09-01264-f002:**
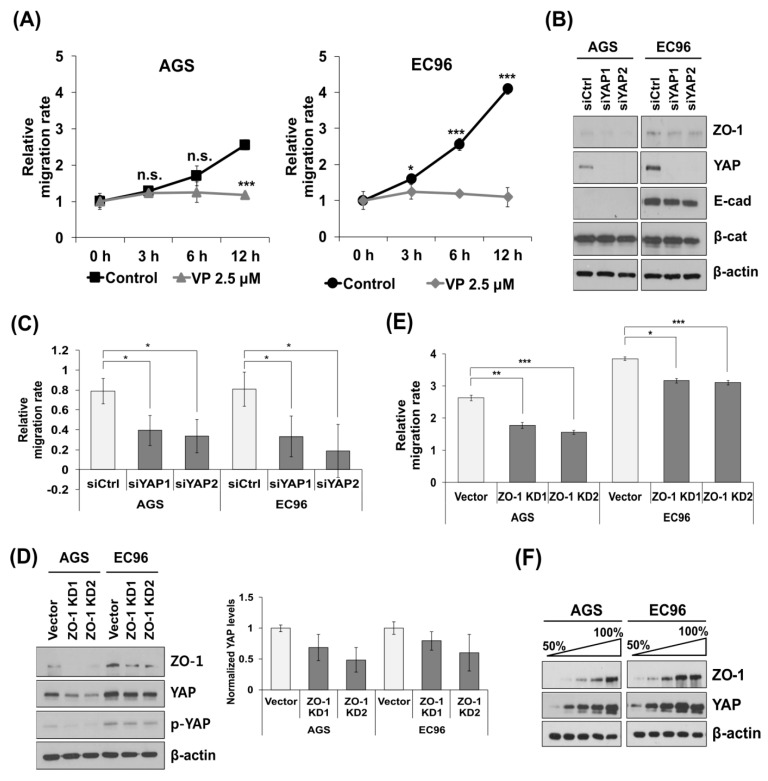
YAP and ZO-1 regulate cell migration. (**A**) AGS and EC96 cells were treated with 2.5 μM verteporfin (VP) for 24 h and subjected to cell migration assay. (**B**,**C**) AGS and EC96 cells were transfected with siYAP RNA and subjected to immunoblot analysis using the indicated antibodies (**B**) or cell migration assay (**C**). (**D**,**E**) AGS and EC96 ZO-1 KD cells were established from AGS and EC96 cells using a lentivirus containing shRNA targeting ZO-1. These KD cells were subjected to immunoblot analysis (**D**) and cell migration assay (**E**). Quantification using densitometry of triplicate repeats normalized to β-actin was performed for YAP expression (±SD, *n* = 3). (**F**) AGS and EC96 cells were cultivated in different confluences and subjected to immunoblot analysis. ZO-1 and YAP expression was increased in more confluent cultures. n.s. = not significant, * *p* < 0.05, ** *p* < 0.01 and *** *p* < 0.001.

**Figure 3 biomedicines-09-01264-f003:**
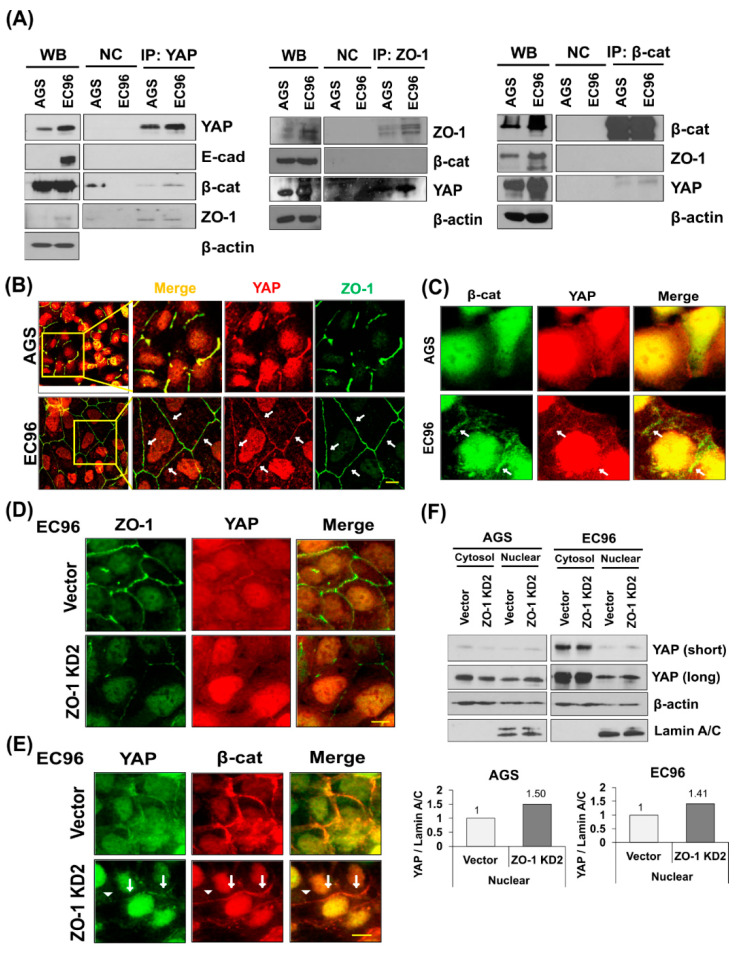
YAP is associated with ZO-1 at TJs. (**A**) IP was performed on AGS and EC96 cell lysates using anti-YAP, -ZO-1, or -β-catenin antibodies. Precipitates were subjected to immunoblot analysis using the indicated antibodies. (**B**,**C**) AGS and EC96 cells were subjected to IF analysis to evaluate the interaction between YAP and ZO-1 (**B**), or YAP and β-catenin (**C**). Arrows indicated regions of colocalization. Scale bar = 10 μm. (**D**,**E**) EC96 ZO-1 KD cells were subjected to IF analysis to evaluate the interaction between YAP and ZO-1 (**D**), or YAP and β-catenin (**E**). Arrows indicated regions of colocalization. Arrowheads indicated regions of non-colocalization. Scale bar = 10 μm. (**F**) EC96 ZO-1 KD cells were subjected to subcellular fractionation analyses to evaluate nuclear YAP expression. Short and long exposures of the blot are shown. Densitometry analyses were performed to evaluate the YAP/Lamin A/C ratio in AGS and EC96 cells.

**Figure 4 biomedicines-09-01264-f004:**
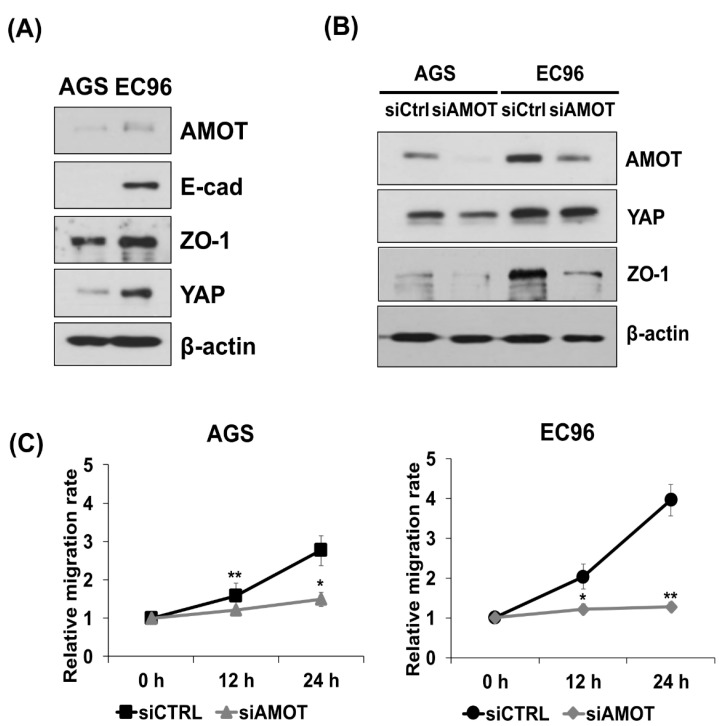
AMOT mediates the linkage between YAP and ZO-1. (**A**) AGS and EC96 cells were subjected to immunoblot analysis using the indicated antibodies. (**B**,**C**) AGS and EC96 cells transfected with siAMOT RNA were subjected to immunoblot analysis using the indicated antibodies (**B**) and cell migration assay (**C**). * *p* < 0.05 and ** *p* < 0.01 in (**C**).

**Figure 5 biomedicines-09-01264-f005:**
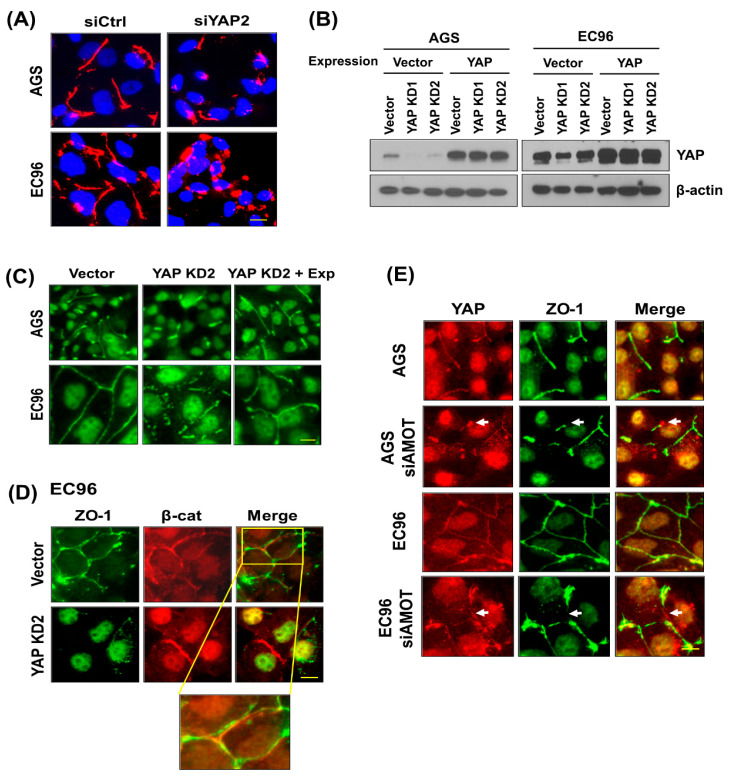
YAP is required for maintenance of ZO-1-containing TJ integrity. (**A**) AGS and EC96 cells were transfected with siYAP RNA and subjected to IF analysis using an anti-ZO-1 antibody and Hoechst dye. Scale bar = 10 μm. (**B**) YAP KD cells and YAP KD cells transfected with YAP cDNA to restore YAP expression were subjected to immunoblot analysis using an anti-YAP antibody. β-actin was used as the protein loading control. (**C**) YAP KD cells and YAP-restored YAP KD cells (YAP KD + Exp) were subjected to IF analysis using an anti-ZO-1 antibody. Scale bar = 10 μm. (**D**) Control and YAP KD EC96 cells were subjected to IF analysis using an anti-β-catenin antibody. Scale bar = 10 μm. (**E**) AGS and EC96 cells were transfected with siAMOT RNA and subjected to IF analysis using an anti-ZO-1 or -YAP antibodies. Arrows indicated dissociation of YAP and ZO-1. Scale bar = 10 μm.

**Figure 6 biomedicines-09-01264-f006:**
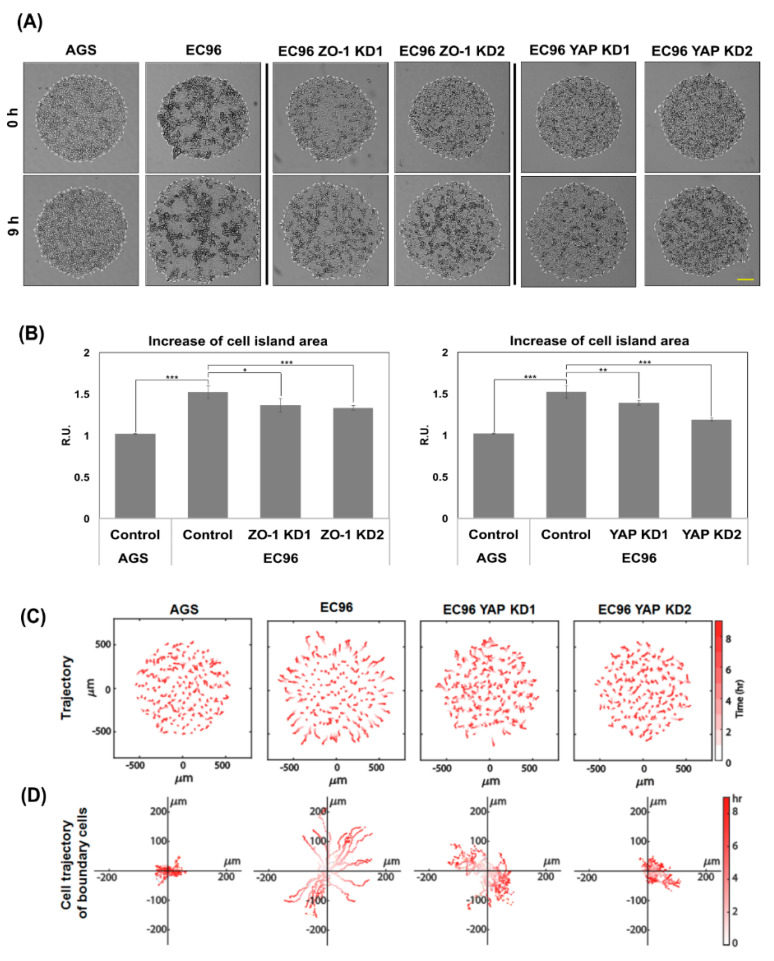
YAP participated in straightforward cell migration of EC96. (**A**) AGS, EC96, EC96 ZO-1 KD, and YAP KD cells were subjected to cell island expansion analysis. EC96 YAP KD cells were established by introducing YAP shRNA to EC96 cells. Bright-field images of cell islands at 0 h and 9 h are shown. Scale bar = 200 μm. (**B**) Quantification of cell island expansion; means of triplicate experiments. * *p* < 0.05, ** *p* < 0.01 and *** *p* < 0.001. (**C**) Trajectory of cells in AGS, EC96, and EC96 YAP KD cell islands. Red color intensity indicates time elapsed. (**D**) Trajectories from the initial locations of boundary cells in AGS, EC96, and EC96 YAP KD cell islands. Red color intensity indicates time elapsed.

## Data Availability

Data is contained within the article or Appendix A.

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
