# Peer review of "Yes-Associated Protein Is Required for ZO-1-Mediated Tight-Junction Integrity and Cell Migration in E-Cadherin-Restored AGS Gastric Cancer Cells"

_biomedicines, 2021, doi:10.3390/biomedicines9091264_

Round 1

Reviewer 1 Report

The authors presented an interesting research on the role of YAP and ZO-1 in cancer cell migration. The topic of the manuscript is full of interest due the need of identifying molecules that may play a role in tumor progression and metastasis. The pathway described may be result in a possible target to identify tumors more prone to metastasis.

I congratulate to the authors for the study and experiments performed.

The manuscript is well written and fully intelligible. However, compared to introduction, discussion appears less comprehensive and more details in similar results may be useful to understand the power of these results.

Moreover, this is an in-vitro study and all results presented are related to the cell cultures. Despite it will not influence the force of results, I suggest to indicate in discussion that the nature of the study need further in-vivo study to better understand the mechanism described.

Author Response

The authors presented an interesting research on the role of YAP and ZO-1 in cancer cell migration. The topic of the manuscript is full of interest due the need of identifying molecules that may play a role in tumor progression and metastasis. The pathway described may be result in a possible target to identify tumors more prone to metastasis.

I congratulate to the authors for the study and experiments performed.

The manuscript is well written and fully intelligible. However, compared to introduction, discussion appears less comprehensive and more details in similar results may be useful to understand the power of these results.

Moreover, this is an in-vitro study and all results presented are related to the cell cultures. Despite it will not influence the force of results, I suggest to indicate in discussion that the nature of the study need further in-vivo study to better understand the mechanism described.

Response: According to reviewer’s comment, we added the sentence as follow in line 357-358, in discussion.

“In addition, these cell culture-based experiments required to expand in vivo experiments for further verification.”

Reviewer 2 Report

This manuscript by Kim et al investigates the cross-regulation between YAP, ZO-1 in gastric cancer cells that don’t express (AGS) or express E-cadherin (EC96) and the effects of this cross-regulation on cell migration. This is an interesting work and topic, since recent findings in the field have revealed that cell-cell adhesion components, such as E-cadherin or ZO-1, promote collective cell migration, contrary to the previously held notion that these proteins inhibit it. The manuscript is overall well-written and seems promising; however, there are a number of issues, including mis- or over-interpretation of data, missing controls, and poor data quality, which preclude full evaluation of the work as suitable for publication, as presented. In detail:

Major issues:

1) Although the manuscript revolves around the connection between YAP and ZO-1, the authors include NF-kB signaling, without this being well-connected with the rest of the work and with the presented data being thin, at best. For example, in line 160 it is stated that: “nuclear NF-kB was increased in EC96 cells compared to AGS cells 160 (Figure 1F)”. This cannot be concluded by the single IF image, which shows only 4-5 cells and is of poor quality. Either IF quantification and/or fractionation need to be conducted to support any change in NF-kB  activity. Also, the related Fig. S2 is not described at all - there is not even a Figure legend and there is no way to understand what the authors present there. Overall, the NF-kB involvement is not supported and is confusing, as presented.

2) Although p-YAP seems to strongly localize at cell-cell junctions, there is no junctional localization of total YAP in EC96 cells (Fig. 1F) - how is this explained? There cannot be p-YAP without the total. There is a critical discrepancy here that should be addressed. Maybe use of additional YAP antibodies?

3) In lines 158-159, it is stated: “membrane localization of phosphorylated YAP might result in escape from cytoplasmic degradation machinery”. This is speculation, unless it has been published (and a reference is cited here) or proven. Otherwise, it should be removed. Similarly, in lines 313-314, the authors extrapolate that YAP and ZO-1 levels are regulated at the level of protein stability. A cycloheximide experiment would determine whether it is indeed protein stability, otherwise it is again speculation. For example, why not this being miRNA-mediated regulation - or translation efficiency?

4) Immunofluorescence images are of low quality and blurry, in some cases. It is not even certain if these are confocal images, which is required to be able to properly assess subcellular localization. Please provide better resolution images obtained by confocal microscopy. Also, regarding Figure 3B: using cell layers of higher confluency could help better visualize the junctional localizations.

5) In Figures 1A, 2C, 2E, 4C is unclear what the units are on the Y axis - what do they show? Units need to be precisely specified. Also, to exclude the possibility that what is observed in the wound scratch assays is a proliferation effect, these assays should be repeated using a low dose of mitomycin (as per common wound scratch protocols).

6) Western blots need to be quantified in several instances:

-Figure 1B: Total YAP is also upregulated, so is it also phosphorylation increased or just levels? Quantification here is required.

-Figure 2D: Please quantify - only the AGS cells seem to have lower YAP.

-Figure 3F: It is not clear whether there is an increase of nuclear ZO-1 - please quantify. Also, please use GAPDH as cytoplasmic control for the fractionation assays.

7) To establish the YAP-ZO-1-AMOT link, an IF and IP of AMOT kd cells is required, showing that e.g. YAP is not at the junctions and/or its interaction with ZO-1 is affected.

Minor issues:

1) It is “Adherens” junctions - not “adherence”

2) As mentioned above, Figure legends are missing from the Supplemental Figures

Author Response

1) Although the manuscript revolves around the connection between YAP and ZO-1, the authors include NF-kB signaling, without this being well-connected with the rest of the work and with the presented data being thin, at best. For example, in line 160 it is stated that: “nuclear NF-kB was increased in EC96 cells compared to AGS cells 160 (Figure 1F)”. This cannot be concluded by the single IF image, which shows only 4-5 cells and is of poor quality. Either IF quantification and/or fractionation need to be conducted to support any change in NF-kB activity. Also, the related Fig. S2 is not described at all - there is not even a Figure legend and there is no way to understand what the authors present there. Overall, the NF-kB involvement is not supported and is confusing, as presented.

Response 1: Previously, we reported that phenotypes of AGS after E-cadherin re-introduction produced malignant changes including increase of cell proliferation and these changes were mediated by increase NF-kB signaling, as a compensatory mechanism for E-cadherin-mediated suppression of Wnt signaling (Ref. 27). Therefore, we only showed representative IF result for increase of NF-kB signaling in this manuscript (Fig. 1F). To show the involvement of NF-kB signaling further, we tried inhibition of NF-kB signaling using Bay11-7082 and showed reduction of cell migration and expression levels of ZO-1 and YAP of EC96 cells in supplementary figure S2. And we correct the sentence for result from line 169 to 173.

“Consistent with our previous results (Ref. 27), NF-kB signaling appeared to be elevated in EC96 cells as high level of nuclear NF-kB was observed (Figure 1F). Furthermore, inhibition of NF-kB signaling using Bay11-7082 reduced cell migration as well as expressions of ZO-1 and YAP (Figure S2), indicating involvement of NF-kB signaling.”

We regret omission of legend of supplementary figure in former version of this MS (we uploaded legends as separate file). In this revised MS, we completed figure legend in each supplementary figure.

2) Although p-YAP seems to strongly localize at cell-cell junctions, there is no junctional localization of total YAP in EC96 cells (Fig. 1F) - how is this explained? There cannot be p-YAP without the total. There is a critical discrepancy here that should be addressed. Maybe use of additional YAP antibodies?

Response 2: The reason why junctional localization of total YAP appears blurry in Fig. 1F might be associated with Immunofluorescence protocol. Cells seeded on a coverslip were incubated with pretreatment solution for 1 min at room temperature before fixation. Pretreatment solution consists of 0.4% Triton X-100, 0.4% Paraformaldehyde, and 1X PBS. This Triton X-100 pretreatment removed cytosolic soluble proteins and enhance visualization cellular structures including membrane protein and cytoskeleton. In figure 1F, we did not treat the cells with Triton X-100 to visualize cytosolic YAP protein, which make weak membrane protein staining.

In this MS, we changed the previous Fig 1F to more appropriate images in addition to images of pretreated result of YAP.

 Also, we added some comment about this pretreatment process in method section line 138-140.

“For enhanced visualization of junctional proteins, cells were pretreated with 0.4% Triton X-100, 0.4% paraformaldehyde in phosphate buffered saline prior to fixation.”

Also, we confirm that junctional localization of phosphorylated YAP and total YAP using additional YAP antibody. As a result, phospho-serine 127 or –serine 397 of YAP is co-localized with total YAP in EC96 cell-cell junctions. These results were added to supplementary figure 1 (Fig. S1A).

3) In lines 158-159, it is stated: “membrane localization of phosphorylated YAP might result in escape from cytoplasmic degradation machinery”. This is speculation, unless it has been published (and a reference is cited here) or proven. Otherwise, it should be removed. Similarly, in lines 313-314, the authors extrapolate that YAP and ZO-1 levels are regulated at the level of protein stability. A cycloheximide experiment would determine whether it is indeed protein stability, otherwise it is again speculation. For example, why not this being miRNA-mediated regulation - or translation efficiency?

Response 3: As reviewer indicated, we emphasized the speculation too much. We deleted the speculative comments and changed the sentence to

Line 167-169

“Phosphorylated YAP at serine-127 and -397 accumulated at cell-cell junctions (Figure S1), although phosphorylation at serine-397 is linked to ubiquitination and degradation [32].”

Line 323-325

“However, the transcript levels of ZO-1 and YAP were not affected by siYAP RNA transfection and ZO-1 KD, respectively (Figure S3), suggesting of post-transcriptional regulation.”

4) Immunofluorescence images are of low quality and blurry, in some cases. It is not even certain if these are confocal images, which is required to be able to properly assess subcellular localization. Please provide better resolution images obtained by confocal microscopy. Also, regarding Figure 3B: using cell layers of higher confluency could help better visualize the junctional localizations.

Response 4: We changed the figure 3B to show images of more confluent cells. In this changed image showed colocalization of YAP and ZO-1 better.

5) In Figures 1A, 2C, 2E, 4C is unclear what the units are on the Y axis - what do they show? Units need to be precisely specified.

 Response 5: It is our mistake. The Y axis showed that the relative migration rate of the value of control cells against experimental cells. We set the migration rate of control cells as 1, therefore we changed the unit of Y axis in figure 1A.

Also, to exclude the possibility that what is observed in the wound scratch assays is a proliferation effect, these assays should be repeated using a low dose of mitomycin (as per common wound scratch protocols).

Response 5: We also asked how much increased cell proliferation influenced on increase of cell migration. We did not perform migration assay using mitomycin. But to answer this question, we employed single cell-based cell migration assay which allow to monitor individual cell movements (Fig. 6). In figure 6, cell motility within a cell island was analyzed as the number and density of clustered cells were controlled using a pattern of homogeneous size (1 mm) and shape. In trajectory analysis shown in figure 6C and 6D, the cells located in periphery of cell cluster migrate longer paths clearly in comparison to those of inner cells. Similar experiments and verification of results were described before (Ref. 31).

 In addition, we added detailed analysis of the relationship between migration and proliferation in supplementary figure S6. In detail, figure 6A and B showed that cell island area of EC96 cells was increased than that of AGS. These results equate to wound scratch assays on figure 1A. In order to confirm whether the area expansion was due to the cell proliferation, cell number and cell island area were analyzed using figure 6 raw data and the results appended to figure S6.

 Figure S6A is an image of single cell segmentation in a cell island to determine the number of cells on the cell image, and the number of cells was counted according to time change.

After 9h growth, number of EC96 cells was increased 1.64-fold compared to the number of cells initially seeded, but number of AGS cells was increased 1.10-fold (Figure S6B). In cell area expansion analysis, area of EC96 cells was increased 1.73-fold but that of AGS cells was increased only 1.08-fold (Figure S6C). In figure S6D, expanded area of cell island at cultured for 9h was divided by initial cell number and the ratio was presented as graph. EC96 cells showed increase of ratio significantly in comparison to that of AGS cells. Considering trajectory analysis in Fig 6C and D, increase of area was mostly due to increase of cell motility of boundary cells rather than increase proliferation of cells in island because most inner cells is less motile similarly to AGS cells. Altogether our single cell-based analysis might support the results of EC96 cell migration shown in scratch assay.

 And we changed the sentence to

Line 276-278

“The area expansion of EC96 appeared to increase slightly in case of correction by initial cell number (Figure S6). To verify if migration of EC96 indeed elevated, cellular trajectories were analyzed.”

6) Western blots need to be quantified in several instances

-Figure 1B: Total YAP is also upregulated, so is it also phosphorylation increased or just levels? Quantification here is required.

Response 6: We measured the band densities and presented as ratio of phospho-YAP by total YAP. The results showed that the ratio of phospho-YAP was increased in EC96 cells.

-Figure 2D: Please quantify - only the AGS cells seem to have lower YAP.

Response 6: We measured the band densities and presented in figure 2D. In Figure 2D, reduction of YAP in YAP KD EC96 cells appeared not so clear. In supplementary figure S5, similar set of immunoblot analysis showed some more clear reduction of YAP. We believe there are batch differences between experiments and we intended to present same batch for each figure instead of selection of the best quality result from several batches. Therefore, we analyzed the band densities of three independent result of YAP in ZO-1 KD cells and presented in figure 2D. 

-Figure 3F: It is not clear whether there is an increase of nuclear ZO-1 - please quantify. Also, please use GAPDH as cytoplasmic control for the fractionation assays.

Response 6: In figure 3F, the increase of nuclear YAP of EC96 ZO-1 KD2 cells was not clear. To show more clear results, we added the result of long exposure film and we measured density of this long-exposed result and presented in figure 3F.

7) To establish the YAP-ZO-1-AMOT link, an IF and IP of AMOT kd cells is required, showing that e.g. YAP is not at the junctions and/or its interaction with ZO-1 is affected.

Response 7: We tried to establish the AMOT KD cells several times but unfortunately failed. So, we used only siRNA-mediated knockdown cells. But regarding to relationship among three proteins, AMOT-YAP-ZO-1, association of YAP with AMOT is known phenomenon.

In this MS, we tried to describe reciprocal relationship of expression of three proteins in figure 5 using siAMOT transfected cells. We added supplementary figure S5A to show reduction of AMOT expression in ZO-1 KD and YAP KD cells.

We added the sentence in line 234-237

“AGS and EC96 cells were transfected with siAMOT RNA showed reduction of AMOT, YAP, and ZO-1 expression (Figure 4B) and similarly AMOT expression was decreased in ZO-1 KD and YAP KD cells (Figure S5).”

Careful review of our IF results revealed that ZO-1 and YAP staining pattern appeared fragmented and sometimes separation of YAP and ZO-1 staining in AMOT siRNA transfected cells, suggesting of AMOT is involved in linkage of YAP and ZO-1. But YAP appeared to locate still in membrane where YAP was dissociated from ZO-1. Because we enhance the visualization of membrane protein by treatment of triton x-100 before fixation of IF samples, much of cytoplasmic proteins might be removed. Even in absence of triton X-100 treatment, increase of cytoplasmic YAP was hard to determine. If YAP still locate at membrane even where AMOT expression is reduced enough, the mechanism of membrane localization of YAP should be further investigated. Anyways, our results showed that eventually the YAP expression as well as ZO-1 expression seemed to be reduced even at membrane after AMOT siRNA transfection.

Minor issues:

1) It is “Adherens” junctions - not “adherence”

We corrected all misspelling of Adherence to adherens.

2) As mentioned above, Figure legends are missing from the Supplemental Figures

 We added figure legends to all supplementary figures.

Round 2

Reviewer 2 Report

I am happy with the Authors's efforts to address the comments